# Allosteric regulation accompanied by oligomeric state changes of *Trypanosoma brucei* GMP reductase through cystathionine-β-synthase domain

Akira Imamura[1,7], Tetsuya Okada [1,7], Hikaru Mase[1], Takuya Otani[1], Tomoka Kobayashi [1], Manatsu Tamura[1], Bruno Kilunga Kubata[2], Katsuaki Inoue [3], Robert P. Rambo [3], Susumu Uchiyama [4,5], Kentaro Ishii [5,6], Shigenori Nishimura [1] & Takashi Inui [1✉]

Guanosine 5′-monophosphate reductase (GMPR) is involved in the purine salvage pathway and is conserved throughout evolution. Nonetheless, the GMPR of *Trypanosoma brucei* (TbGMPR) includes a unique structure known as the cystathionine-β-synthase (CBS) domain, though the role of this domain is not fully understood. Here, we show that guanine and adenine nucleotides exert positive and negative effects, respectively, on TbGMPR activity by binding allosterically to the CBS domain. The present structural analyses revealed that TbGMPR forms an octamer that shows a transition between relaxed and twisted conformations in the absence and presence of guanine nucleotides, respectively, whereas the TbGMPR octamer dissociates into two tetramers when ATP is available instead of guanine nucleotides. These findings demonstrate that the CBS domain plays a key role in the allosteric regulation of TbGMPR by facilitating the transition of its oligomeric state depending on ligand nucleotide availability.

---

[1] Department of Applied Life Sciences, Graduate School of Life and Environmental Sciences, Osaka Prefecture University, 1-1 Gakuen-cho, Naka-ku, Sakai, Osaka 599-8531, Japan. [2] AU/NEPAD Agency Regional Office for Eastern and Central Africa, Nairobi, Kenya. [3] Diamond Light Source, Harwell Science and Innovation Campus, Didcot, Oxfordshire OX11 0DE, UK. [4] Department of Biotechnology, Graduate School of Engineering, Osaka University, 2-1 Yamadaoka, Suita, Osaka 565-0871, Japan. [5] Exploratory Research Center on Life and Living Systems (ExCELLS), National Institutes of Natural Sciences, 5-1 Higashiyama, Myodaiji-cho, Okazaki 444-8787, Japan. [6] Present address: Department of Biotechnology, Graduate School of Engineering, Osaka University, 2-1 Yamadaoka, Suita, Osaka 565-0871, Japan. [7] These authors contributed equally: Akira Imamura, Tetsuya Okada. ✉email: inuit@bioinfo.osakafu-u.ac.jp

The appropriate regulation of intracellular concentrations of purine nucleotides is critical for the health of cells, and most cells have the mechanism of either synthesizing de novo or salvaging purine nucleotides to maintain their intracellular concentrations; nevertheless, this is not the case for most parasitic protozoa. *Trypanosoma brucei* is one such parasite that causes African trypanosomiasis by infecting to both human and animals, and depends solely on salvaging the purines produced by the host animals; interconversion between adenine and guanine nucleotides is indispensable for this parasite[1–4].

We recently have characterized the enzymes involved in purine nucleotide salvage in *T. brucei*: guanosine 5′-monophosphate reductase (GMPR) and inosine 5′-monophosphate dehydrogenase (IMPDH)[5,6]. The former catalyzes the conversion of guanosine 5′-monophosphate (GMP) to inosine 5′-monophosphate (IMP), whereas the latter utilizes IMP to produce xanthosine 5′-monophosphate. Our previous studies have demonstrated that a purine nucleotide analog, ribavirin 5′-monophosphate, acts as an inhibitor for both *T. brucei* GMPR (TbGMPR) and IMPDH (TbIMPDH), but also shows an anti-trypanosomal effect in culture when provided in the nucleoside form, ribavirin[5,6]. In general, both GMPRs and IMPDHs have strong similarities in amino acid sequence, and their catalytic domains share the common structure of a $(\beta/\alpha)_8$ barrel, also referred to as a TIM barrel[7]; nevertheless, these two enzymes are still distinctive each other by the presence or absence of an additional domain[8]. This additional domain is known as a cystathionine-β-synthase (CBS) or Bateman domain that consists of a tandem repeat of α-β-β-α folds, and is found in IMPDHs of all organisms reported to date. The CBS domain of IMPDHs has been shown to participate in the regulation of its activity and conformation in response to the concentrations of purine nucleotides[8]. Despite of these previous findings, we and others have recently revealed that GMPRs of trypanosomatids, including *Trypanosoma* and *Leishmania* species, uniquely possesses a CBS domain that is absent from the GMPRs of other species[6,9], although the structure of a GMPR harboring a CBS domain still remains undetermined. These observations prompted us to investigate the structure and reaction mechanism of TbGMPR.

In the present study, we investigated the biochemical and structural effects of adenine and guanine nucleotides on TbGMPR. We found that the binding of adenine and guanine nucleotides to the CBS domain have opposing effect on the allosteric regulation of TbGMPR activity. A combination of X-ray crystallography and size-exclusion chromatography small-angle X-ray scattering (SEC-SAXS) analysis clearly revealed that TbGMPR can exist as a tetramer or an octamer depending on the nucleotide species that is bound to the CBS domain. Our findings suggest that the change in the oligomeric state of TbGMPR is responsible for allosteric regulation by nucleotide binding to the CBS domain.

## Results

**Opposite modulation of TbGMPR activity by purine nucleotides.** Enzymes harboring a CBS domain usually change their activities in the presence of purine nucleotides[10–12]; therefore, we sought to examine whether purine nucleotides modify the activity of TbGMPR. In the presence of GTP, the initial velocity of TbGMPR was upregulated in a concentration-dependent manner (Fig. 1a), and the $EC_{50}$ value was estimated to be 4.8 μM. In contrast, TbGMPR exhibited a decrease in its initial velocity in the presence of ATP with an $EC_{50}$ value of 160 μM (Fig. 1b). The effect of each triphosphate nucleotide was maintained when added as a magnesium complex, i.e. Mg-GTP or Mg-ATP (Fig. 1a, b). These results indicate that TbGMPR is positively and negatively regulated by GTP and ATP, respectively, with and without magnesium ions. Kinetic analysis demonstrated that the

initial reaction velocity of TbGMPR without ligand nucleotides showed a sigmoidal curve when plotted against the concentrations of GMP, and the plots were well-fitted to the Hill equation (Fig. 1c, open circles). The kinetic parameters $K_{0.5}$ and $k_{cat}$ values for GMP were determined as $184 \pm 3$ μM and $16.7 \pm 0.15$ min$^{-1}$, respectively (mean ± s.d.; Table 1). The Hill constant ($n_{Hill}$) was calculated to be $3.04 \pm 0.12$, meaning that GMP induced a positive cooperativity effect on TbGMPR. A similar sigmoidal curve was observed using recombinant TbGMPR prepared via an affinity purification with glutathione-*S*-transferase (GST) tag and subsequent tag-removal (Supplementary Fig. 1). These results indicate that GMP induces a positive cooperative effect on TbGMPR, both with and without a terminal tag.

The addition of GTP to the reaction mixture enhanced TbGMPR activity by decreasing the $K_{0.5}$ accompanied with the increase of the $k_{cat}$ value in a concentration-dependent manner (Table 1, Fig. 1c and Supplementary Fig. 2). The $n_{Hill}$ value in the presence of 1 mM GTP was $1.00 \pm 0.34$. In contrast, ATP showed an inhibitory effect on TbGMPR activity by lowering the substrate–enzyme affinity as indicated by the $K_{0.5}$ value of $1200 \pm 12$ μM, while the $k_{cat}$ and $n_{Hill}$ values were relatively unchanged compared to those obtained in the absence of ligands (Table 1, Fig. 1c and Supplementary Fig. 2). The increase in the $K_{0.5}$ value was observed in an ATP concentration-dependent manner (Supplementary Fig. 2a). The opposed effects of GTP and ATP on TbGMPR activity were clearly observed when the values of catalytic efficiency ($k_{cat}/K_{0.5}$) were plotted against each ligand concentration. Although the $k_{cat}/K_{0.5}$ value of TbGMPR in the absence of the ligands was determined as $0.913 \times 10^5$ M$^{-1}$ min$^{-1}$, it was increased by the addition of 10 μM GTP, and finally showed $4.36 \times 10^5$ M$^{-1}$ min$^{-1}$ with 1 mM GTP (Fig. 1d and Table 1). In contrary, the $k_{cat}/K_{0.5}$ value was decreased by the addition of ATP and reached to $0.111 \times 10^5$ M$^{-1}$ min$^{-1}$ with 1 mM ATP (Fig. 1d and Table 1). These results indicate that guanine and adenine nucleotides are the allosteric regulators responsible for the positive and negative regulation of TbGMPR activity, respectively, by altering the affinity of the substrate GMP for the reaction center.

To investigate the regulation of TbGMPR activity by purine nucleotides under the physiological conditions, we examined whether GTP can activate the ATP-inactivated enzyme by measuring the activities in the presence of both 1 mM ATP and various concentrations of GTP. At the concentrations of 10 μM or less, GTP was ineffective on ATP-inactivated enzyme, however, 100 μM GTP was sufficient to revoke the inhibitory effect of ATP (Fig. 1e). The $n_{Hill}$ value in the latter condition was calculated to be $1.03 \pm 0.68$; this was very close to the value in the presence of GTP alone rather than that without the ligands. Inversely, the addition of 3 mM ATP inhibited the 100 μM GTP-activated TbGMPR, whereas 1 mM ATP showed only a moderate effect (Supplementary Fig. 3a). ATP was less effective to 1 mM GTP-activated TbGMPR (Supplementary Fig. 3b). These data indicate that TbGMPR is reversibly regulated between the activated and inhibited state by binding of GTP and ATP, respectively, and this state transition is dependent to the concentration ratio of these nucleotide ligands.

**Allosteric binding of purine nucleotides to CBS domain.** We investigated the binding of ligand nucleotides to the CBS domain of TbGMPR by fluorescence quenching of the tryptophan residues. TbGMPR has two tryptophan residues that are located in its CBS domain (Trp115 and Trp120). To avoid interference between the two tryptophan residues, we replaced Trp115 with arginine (TbGMPR W115R). The enzymatic properties of TbGMPR W115R were almost identical to those of the wild-type

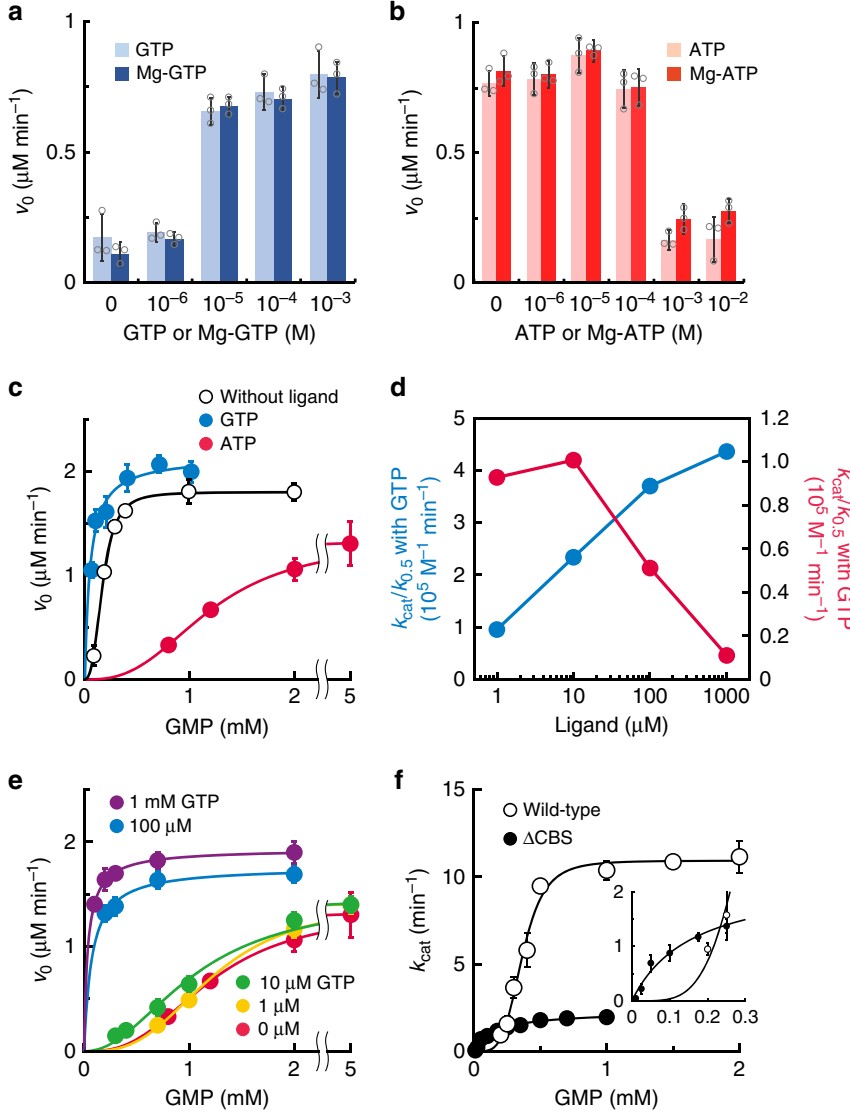

**Fig. 1 Kinetic analysis of TbGMPR in the presence of purine nucleotides. a**, **b** The initial velocities of TbGMPR in the presence of GTP (**a**) or ATP (**b**) at a fixed concentration of GMP and NADPH. Each trinucleotide was used as a premix with (*solid bars*) or without (*shaded bars*) equivalent amount of magnesium ions. **c** The initial velocities of TbGMPR were plotted against the concentrations of GMP in the absence (open circles) or presence of 1 mM GTP (blue) or ATP (red) at a fixed concentration of NADPH. Data were fitted to the Hill equation, as described in the "Methods". **d** The catalytic efficiency ($k_{cat}/K_{0.5}$) of TbGMPR in the presence of various concentrations of GTP (blue) or ATP (red) ligand alone. **e** The initial velocities of TbGMPR in the absence (red) or presence of 1 (orange), 10 (green), 100 μM (blue), or 1 mM GTP (magenta) under fixed concentration of ATP at 1 mM. Note that the kinetics with GTP at 100 μM and 1 mM showed a Michaelis–Menten-like profile. **f** The $k_{cat}$ values of TbGMPR (*open circles*) and TbGMPRΔCBS (*closed circles*) were plotted against the concentrations of GMP. The *inset* shows the data at low concentrations of GMP. Data were obtained from three independent experiments ($n = 3$). Each data point represents a mean ± s.d. in error bars. Source data are provided as a Source Data file.

**Table 1 Kinetic data for TbGMPR wild-type (WT) and TbGMPRΔCBS.**

| TbGMPR | Ligand | $K_{0.5}$ (μM) | $k_{cat}$ (min$^{-1}$) | $k_{cat}/K_{0.5}$ (10$^5$ M$^{-1}$ min$^{-1}$) | $n_{Hill}$ |
|---|---|---|---|---|---|
| WT | None | 184 ± 3 | 16.7 ± 0.15 | 0.913 | 3.04 ± 0.12 |
| WT | GTP | 49.4 ± 8.9 | 21.5 ± 1.68 | 4.36 | 1.00 ± 0.34 |
| WT | ATP | 1200 ± 12 | 13.4 ± 0.10 | 0.111 | 2.72 ± 0.07 |
| ΔCBS | None | 162 ± 36 | 2.29 ± 0.20 | 0.141 | 1.00 ± 0.14 |

enzyme (Supplementary Fig. 4). In the presence of ligand nucleotides, TbGMPR W115R clearly showed fluorescence quenching of Trp120 in a ligand concentration-dependent manner (Fig. 2, upper panels). We also observed blue-shifting of the emission spectra of the protein with either GTP or GMP, but not

with ATP, in a nucleotide concentration-dependent manner. These data suggest that the local environment around Trp120 is rendered more hydrophobic in the presence of guanine nucleotides. When plotting the peak fluorescence intensities against the ratios of ligand to enzyme concentrations, the data for each ligand

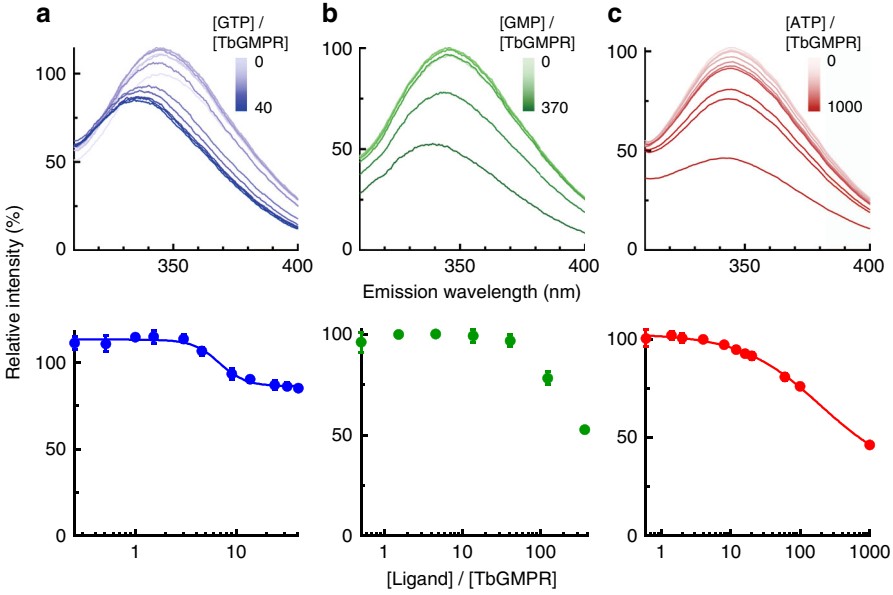

**Fig. 2 Tryptophan fluorescence quenching of TbGMPR W115R by purine nucleotides. a–c** (*Upper panels*) The fluorescence emission spectra of Trp120 in TbGMPR W115R. GTP (**a**), GMP (**b**), and ATP (**c**) were used as ligands. **a–c** (*Lower panels*) The peak value of each plot was re-plotted against the ratio of the ligand to protein concentration shown in logarithmic scale. Plots were fitted to the Hill equation. Data were obtained from three independent experiments (*n* = 3). Each data point represents a mean ± s.d. in error bars. Source data are provided as a Source Data file.

except GMP showed a curve-fitting to the Hill equation (Fig. 2, lower panels), and the dissociation constants for GTP and ATP were calculated as $6.75 \pm 0.58$ and $194 \pm 45\,\mu M$, respectively. Supplementation of magnesium ions showed little or no effect on the spectrum of TbGMPR with GTP or ATP (Supplementary Fig. 5). These results indicate that the purine nucleotides used here bind close to Trp120 in the CBS domain of TbGMPR, and the bindings are independent of magnesium ions.

**Substrate-induced positive cooperativity via CBS domain.** We generated a mutant version of the TbGMPR protein lacking the CBS domain (TbGMPRΔCBS) and used this altered protein to elucidate the involvement of the CBS domain in purine nucleotide-dependent regulation of TbGMPR. TbGMPRΔCBS showed a marked decrease in overall activity, exhibiting a fivefold decrease in $k_{cat}$ value compared to that of the wild-type (Fig. 1f and Table 1). However, the $K_{0.5}$ values of the wild-type and TbGMPRΔCBS were comparable, and the $n_{Hill}$ for TbGMPRΔCBS with GMP was $1.00 \pm 0.14$. These results indicate that deletion of the CBS domain eliminates the positive cooperative effect of GMP on TbGMPR activity while having little effect on the affinity for substrate binding to the catalytic center. In addition, the addition of purine nucleotides other than GMP had no effect on the activity of TbGMPRΔCBS (Supplementary Fig. 6).

**TbGMPR architecture.** To obtain direct evidence for the allosteric binding of purine nucleotides to the CBS domain of TbGMPR, we attempted crystallization, with or without purine nucleotides, of the inactive mutant TbGMPR C318A, in which the catalytic residue Cys318[6] was replaced by an alanine. We obtained the crystal without ligand (C318A apo) and successfully determined the structure, at 2.80-Å resolution, of a GMPR harboring a CBS domain (Supplementary Table 1). A monomer of C318A apo comprised two domains: the catalytic domain (Ser2–Phe97 and Arg226–Gly484) and the CBS domain (Leu98–Ser225). The former displayed a typical TIM-barrel fold with accessory α-helices and antiparallel β-sheets, whereas the latter had a tandem repeat

of α–β–β–α folds characteristic to all CBS domains known to date. PISA data analysis[13] revealed that a tetramer (referring to as tetramer 1 or 2) was formed from four subunits (subunits A–D or A′–D′) related by a fourfold axis, and that the two tetramers were related by a twofold axis perpendicular to the fourfold axis to constitute an octamer (Fig. 3a). The interactions between the adjacent subunits in the tetramer involved a large number of hydrogen bonds and hydrophobic interactions, whereas the formation of the octamer was stabilized by a hydrogen bond and 14 C–C contacts between the CBS domains of subunit A in tetramer 1 and subunit A′ in tetramer 2, and equivalent interactions between subunit B and B′, C and C′, and D and D′ (Supplementary Tables 2 and 3). However, no interaction was observed between the catalytic domains, indicating that the CBS domain is essential for the formation of the TbGMPR octamer.

We also determined the structure of TbGMPR C318A with substrate GMP (C318A/GMP) at 1.90-Å resolution and found that C318A/GMP adopted an octameric structure as observed in C318A apo (Fig. 3b and Supplementary Tables 1–3). However, two tetramers in the C318A/GMP octamer were arranged in a twisted position around the fourfold axis compared to C318A apo, and the overall structure of the C318A/GMP octamer was compressed along the fourfold axis. Thus, it appears that GMP induces the transformation of TbGMPR from a "relaxed" (C318A apo) to a "twisted" form (C318A/GMP). Two GMP molecules were observed in each subunit of C318A/GMP; one was found in the active site, whereas the other was bound to the cleft between the catalytic and CBS domains (Fig. 4a). The interactions between the active site and the substrate GMP molecule involved 12 hydrogen bonds and 7 C–C contacts (Supplementary Fig. 7 and Supplementary Table 4). The interactions between the base moiety of GMP and three amino acid residues (Met401, Ala402, and Glu428) in the catalytic domain appeared to stabilize the α-helix and a portion of the following loop structure that were disordered in C318A apo (Fig. 4b, c). The conformation induced upon substrate binding is considered to have a key role in GMPR activity.

The ligand GMP molecule at the allosteric regulatory site formed 10 hydrogen bonds to and exhibited 13 hydrophobic

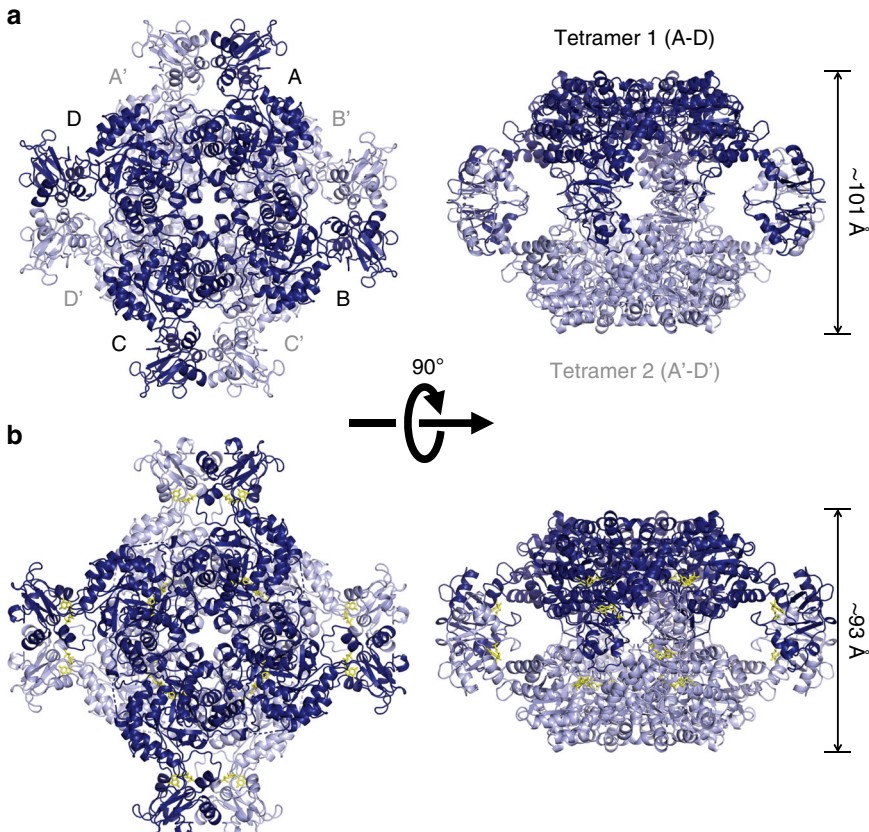

**Fig. 3 Crystal structures of TbGMPR C318A with or without GMP.** Cartoon representation of the octameric structures of C318A apo (relaxed form, **a**) and C318A/GMP (twisted form, **b**), each of which are composed of two tetramers. Tetramer 1 (subunits A–D) and tetramer 2 (subunits A′–D′) are colored in light and dark blue, respectively. GMP molecules are shown as yellow stick representations. Note that C318A/GMP (**b**; *right panel*) is compressed compared to C318A apo (**a**; *right panel*).

interactions with the amino acid residues in the CBS domain; at the same time, the O6 and N7 atoms in the GMP molecule interacted with Arg93 in the catalytic domain via hydrogen bonds (Fig. 4e and Supplementary Table 5). The structures of the isolated catalytic and CBS domains in C318A/GMP could be well-superposed with the respective domains in C318A apo (Supplementary Fig. 8). However, the relative orientation of these domains was obviously different between C318A/GMP and C318A apo: the CBS domain is rotated ~40° between these two structures (Fig. 4a), a change that may be induced by an interaction between the base moiety of GMP and Arg93 (Fig. 4d, e). This hinge motion is inferred to contribute to the transformation from the relaxed to the twisted octamer. We further determined the structure of the wild-type TbGMPR complexed with GTP (TbGMPR/GTP) at 2.50-Å resolution (Supplementary Fig. 9 and Supplementary Table 1). In this structure, single GTP molecules were found only at the allosteric regulatory site on each subunit of TbGMPR/GTP, indicating that GTP acts only as a ligand. The overall structure was almost identical to that of C318A/GMP (RMSD = 0.66 Å for $C_\alpha$ atoms) (Supplementary Fig. 9b). The hinge motion in each subunit and the twisted conformation of the octamer upon allosteric binding of the ligand were also observed in TbGMPR/GTP (Supplementary Fig. 9b, c).

**Regulation of oligomeric state by ligand nucleotide binding**. It is important to study the structure of the protein not only in crystal but also in solution to understand its native structure in a physiological environment. We employed SEC-SAXS analysis to evaluate the oligomeric state and the conformation of TbGMPR

with and without ligands (Fig. 5a). In the absence of ligands, the scattering curve showed the local minimum at a $q$ value of approximately 0.07 Å$^{-1}$; however, the addition of GMP or GTP shifted the valley toward large $q$ values. Meanwhile, the addition of ATP yielded a curve with no valley, a profile that was clearly distinct from those of the other curves. Kratky plots of TbGMPR with and without ligands exhibited a peak at Guinier–Kratky point in all measurements, but except for ATP, indicating that TbGMPR substantially adopts the globular conformation (Fig. 5b). However, the plot of TbGMPR in the presence of ATP showed a peak-shift from Guinier–Kratky point, and had no valley and diverged tailing at higher $q \times R_g$ region. These observations signify that ATP-binding results in an increase of the flexible structure in TbGMPR, leading the subunit conformation into a less compact form. The calculated radii of gyrations ($R_g$), maximum dimensions ($D_{max}$), and molecular masses are summarized in Tables 2 and 3. Porod volumes and $I(0)$ values are shown in Supplementary Table 6. We also carried out SEC-SAXS analysis for TbGMPRΔCBS, so as to investigate the effect on the oligomer formation of deletion of the CBS domain (Table 3 and Supplementary Figs. 10 and 11). The molecular mass of TbGMPRΔCBS calculated from the scattering curve was 137 kDa, which was comparable to the theoretical mass of its tetramer (TbGMPRΔCBS monomer, 39.2 kDa; Supplementary Fig. 12). These results indicate that the CBS domain in TbGMPR is essential for the octamerization of TbGMPR, and therefore, deletion of the CBS domain presumably prevents the interaction between adjacent tetramers.

In order to evaluate the oligomeric states and the conformations of TbGMPR in solution with or without ligands, we applied

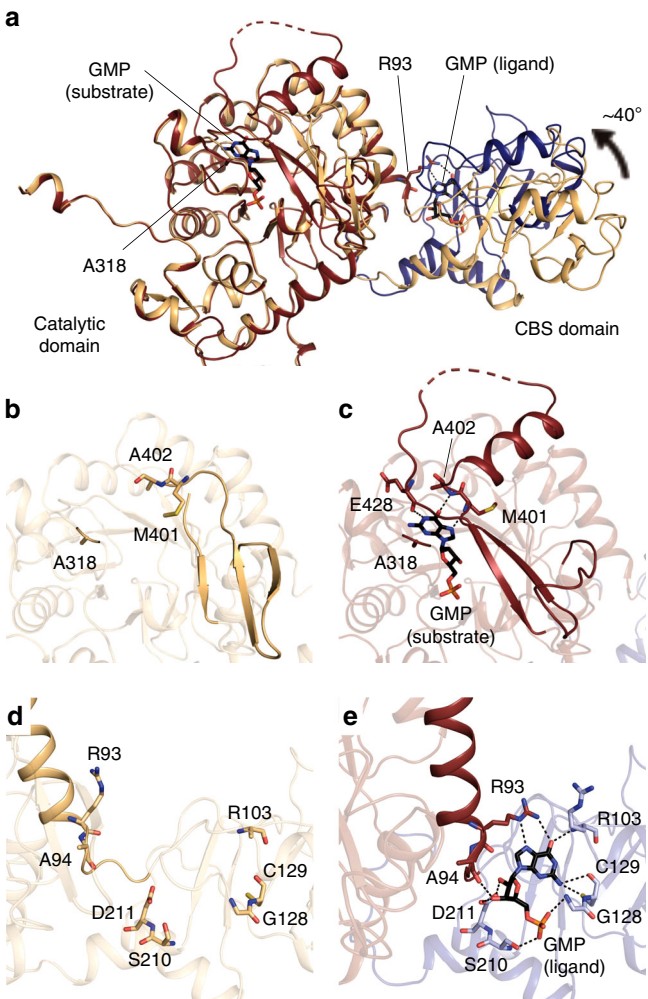

**Fig. 4 Conformational changes in subunits of C318A/GMP and C318A apo. a** Cartoon representation of C318A/GMP and C318A apo subunits with the catalytic domains superimposed. The catalytic and CBS domains of C318A/GMP are colored in red and blue, respectively, and the overall structure of C318A apo is shown in orange. **b–e** Enlarged views of the active site (**b**, **c**) and the allosteric regulatory site (**d**, **e**) of C318A apo (**b**, **d**) and C318A/GMP (**c**, **e**). The hydrogen bonds are depicted as *dashed lines*. Note that Arg93 on the α-helix of the catalytic domain forms hydrogen bonds with the GMP at the allosteric regulatory site.

the program OLIGOMER to obtain fitting curves for the experimental SAXS data based on a linear combination of the theoretical scattering curves of three conformations. Two of the three conformations were the crystal structures of C318A apo and C318A/GMP that represent the relaxed and the twisted octamers, respectively; the third conformation was a tetramer structure generated by dividing the C318A/GMP octamer in half. The OLIGOMER analysis demonstrated that TbGMPR without purine nucleotides was found only as an octamer, though the proportion of the relaxed and twisted forms were estimated as 53.2% and 46.8%, respectively, whereas TbGMPR in the presence of guanine nucleotides was found almost completely as the twisted form (Table 3). In the presence of ATP, however, 60.8% of TbGMPR was found as the tetramer, with the rest of the protein present in the relaxed (28.0%) and twisted (11.2%) octamer conformations. These results suggest that changes in the oligomeric state contribute to the allosteric regulation of TbGMPR by purine nucleotides.

## Discussion

Here, we demonstrate the determination of the crystal structures of *T. brucei* GMPR; a protein that, unlike the GMPRs of host animals, possesses a CBS domain. We further show that TbGMPR activity is regulated by purine nucleotide binding to an allosteric regulatory site formed at the cleft between the catalytic and CBS domains of the enzyme. Recently, the allosteric regulation of GMPR with purine nucleotides was reported in *Leishmania major*[9], an organism that belongs to the same taxon as *T. brucei*. Our study extends the previous findings in *L. major* by demonstrating that allosteric binding of purine nucleotides to TbGMPR triggers drastic conformational changes, an insight that we believe provides crucial information for understanding the mechanism of allosteric regulation of GMPR.

In general, the regulative enzymes with CBS domains possess purine nucleotide binding sites in the domain, and these sites are referred to as canonical binding sites defined by the numbering of the highly conserved Asp residues that interact with the ribose moiety of the nucleotides[12,14,15]. These Asp residues in TbGMPR are Asp149 and Asp211, corresponding to the canonical binding site 1 and 2, respectively. The present crystallographic study of TbGMPR shows that both GMP and GTP share the canonical site 2 on the CBS domain, whereas nothing was observed on the canonical site 1 (Fig. 4, Supplementary Fig. 9 and Supplementary Table 5). However, the affinity of GTP to the canonical site 2 was higher than that of GMP as observed in the fluorescence quenching assay (Fig. 2). These findings indicate that, in the presence of sufficient concentration of GTP, GMP no longer activates TbGMPR since the canonical site 2 has been occupied by GTP. Consequently, the kinetics of TbGMPR in the presence of GTP shows a Michaelis–Menten-like profile with a decrease of the $n_{Hill}$ value to ~1 (Fig. 1c and Table 1).

Crystal structures of GMPRs complexed with GMP have been determined for proteins from *Homo sapiens* (PDBIDs: 2BLE and 2BWG for type-1, and 2A7R for type-2 GMPRs, respectively) and *Bacillus anthracis* (2A1Y); in all cases, the proteins have been found as tetramers[16]. Consistent with those results, the *L. major* GMPR has been reported to exist as a tetramer, as judged by rate zonal centrifugation on a glycerol gradient[9]. However, our crystallographic study and SAXS analysis clearly showed that TbGMPR, with or without guanine nucleotides, rendered to form an octamer composed of a pair of the tetramers interacting via adjacent CBS domains. Furthermore, we demonstrated that the "relaxed" conformation observed in the apo-form of the TbGMPR octamer is transformed into a "twisted" conformation with a slight decrease in size when guanine nucleotides bind to the allosteric regulatory site (Fig. 3, Table 3 and Supplementary Fig. 9). Thus, it is likely that these conformational changes underlie the enzyme activation in the presence of guanine nucleotides, as observed in our kinetic analysis (Fig. 1c, d and Table 1). Current studies of TbGMPRΔCBS demonstrated that deletion of the CBS domain results in a negative effect on the enzyme kinetics, while interfering with the conformational change from TbGMPRΔCBS tetramer to octamer (Fig. 1f and Tables 1–3). CBS domain provides a majority of amino acid residues that participate in the allosteric binding of guanine nucleotides; therefore, it was naturally that TbGMPRΔCBS lost the regulation by the nucleotide ligands (Supplementary Fig. 6). Furthermore, the interaction between the two tetramers consisting of wild-type TbGMPR octamer was maintained only by the amino acid residues on each CBS domain (Supplementary Table 3). These findings indicate that the CBS domain is required for the formation of the octamer, and suggest that the octamer formation through the CBS domain is necessary for TbGMPR activation in the presence of guanine nucleotides. On the other hand, SAXS analysis showed that ATP allows the TbGMPR

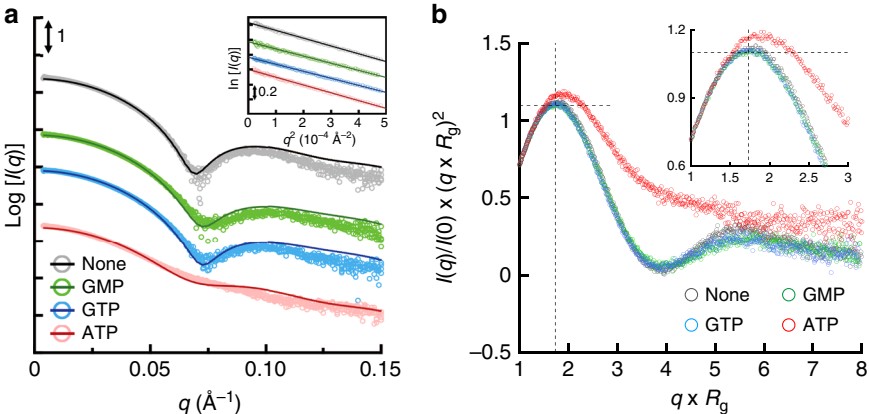

**Fig. 5 SEC-SAXS analysis of TbGMPR in the presence of purine nucleotides. a** SAXS intensities for TbGMPR in the absence (black) or presence of GMP (green), GTP (blue), or ATP (red) were plotted against the scattering vector $q$ defined as $4\pi\sin\theta/\lambda$. The scattering intensity $I(q)$ was represented in absolute intensity. The fitting curves were calculated by OLIGOMER. The *inset* shows the Guinier plots of each curve. **b** Dimensionless Kratky plots of TbGMPR in the presence of purine nucleotides. The plots were calculated for TbGMPR in the absence (gray) or presence of GMP (green), GTP (blue), or ATP (red). Cross-hair marks the Guinier–Kratky point (1.732, 1.1), the main peak position for globular particles. The *inset* shows the enlarged view of the peak position. Data were obtained from three independent experiments ($n = 3$). Source data are provided as a Source Data file.

**Table 2 SEC-SAXS data for TbGMPR WT and TbGMPRΔCBS.**

| TbGMPR | Ligand | $R_{g,\ Guinier}$[a] (Å) | $R_{g,\ P(r)}$[b] (Å) | $D_{max}$ (Å) |
|---|---|---|---|---|
| WT | None | 55.4 ± 0.56 | 55.4 ± 2.59 | 210.0 |
| WT | GMP | 52.9 ± 0.56 | 53.2 ± 3.32 | 209.5 |
| WT | GTP | 52.8 ± 0.41 | 52.4 ± 2.12 | 210.5 |
| WT | ATP | 55.3 ± 0.68 | 55.7 ± 3.48 | 223.5 |
| ΔCBS | None | 38.2 ± 0.18 | 37.8 ± 0.72 | 145.0 |

[a]$R_g$ values calculated from Guinier plots.
[b]$R_g$ values calculated from $P(r)$ profile data.

octamer to dissociate into two tetramers (Table 3), an event that coincides with the inhibition of enzyme activity. Although the mechanism whereby ATP interferes with octamer formation remains to be elucidated, further crystallographic studies of ATP-bound TbGMPR might provide a solution.

In terms of the structures, TbGMPR shows very high homologies to IMPDHs, a $(\beta/\alpha)_8$ barrel protein also composed of catalytic and CBS domains[7]. The activity regulation of IMPDHs is accompanied by their conformational changes, and the relationship between the activity and the conformation is extensively investigated to date[14,15,17]. In the presence of ATP, IMPDHs form an active octamer composed of two tetramers. Whereas, the allosteric binding of guanine nucleotides alters the conformation of the IMPDH octamers to a compact form, and inhibits the activity by the interaction of the finger domains, the motifs that are believed to be essential for the catalysis. On the other hand, our present findings indicate that TbGMPR activity is regulated in positive and negative by allosteric binding of guanine and adenine nucleotides to the CBS domain, respectively, that is assumed to involve the conformational changes distinct from those observed in IMPDHs; represented as the conversion of relaxed and twisted octamer conformations in the absence and presence of guanine nucleotides, respectively, and the dissociation of the octamer into two tetramers in the presence of ATP. Further investigation of the precise structure of the active center of TbGMPR is necessary to extend the understanding of the allosteric regulation mechanism at structure levels.

In view of the metabolic pathways employed by *T. brucei* (Supplementary Fig. 13), it is logical that TbGMPR activity is regulated in positive and negative fashions by guanine and adenine nucleotides, respectively. That is, TbGMPR forms the twisted octamer to utilize guanine nucleotides when such nucleotides exist in excess; in contrast, adenine nucleotides suppress TbGMPR activity by interfering with octamer formation when adenine nucleotides are in excess, a process resembling feedback regulation. As trypanosomes actively proliferate and move around in the host bloodstream by means of extreme wave beating, these parasites are expected to consume a great deal of energy, which would take the form of intracellular ATP[18,19]. Nevertheless, trypanosomes depend almost completely on glycolysis to produce ATP[20,21], and furthermore, have no machinery for de novo synthesis of purine nucleotides from amino acids and sugars[1–4]. These disadvantages for survival of trypanosomes are expected to force these organisms to strengthen the regulation of machinery for the interconversion between adenine and guanine nucleotides. This requirement might explain why the GMPRs of trypanosomatids harbor the CBS domain, and suggest why TbGMPR activity is controlled in opposing fashions by the binding of adenine and guanine nucleotides to the CBS domain. The intracellular concentrations of GTP and ATP, that are reported to be 1.28 and 3.7 mM, respectively, in *T. brucei* bloodstream forms during logarithmic phase[22], are sufficient to bind to the allosteric regulatory site of TbGMPR. Our kinetic study revealed that only 100 µM GTP was enough to re-activate TbGMPR inactivated by 1 mM ATP (Fig. 1e), and furthermore, the dissociation constants determined here by fluorescence quenching experiments indicate that the affinity of GTP is 30-fold higher than that of ATP (Fig. 2). These findings suggest that TbGMPR in trypanosomes ordinarily forms active and twisted octamer under the physiological conditions, enabling to maintain the intracellular ATP at higher concentration than GTP.

In conclusion, we have determined the crystal structure of *T. brucei* GMPR that harbors a CBS domain uniquely found in GMPRs of trypanosomatids. Our kinetic and structural analyses clearly demonstrate that the CBS domain has a pivotal role in the allosteric regulation of the activity and structure of TbGMPR, facilitating the transitions among three conformations: a tetramer, and octamers with relaxed and twisted conformations. These conformational changes are directed by binding of either adenine

| TbGMPR | Ligand | Mass (kDa) | Tetramer (%) | Octamer, relaxed (%) | Octamer, twisted (%) |
|---|---|---|---|---|---|
| WT | None | 494 | 0 | 53.2 | 46.8 |
| WT | GMP | 453 | 2.2 | 0 | 97.8 |
| WT | GTP | 462 | 0 | 0 | 100 |
| WT | ATP | 314 | 60.8 | 28.0 | 11.2 |
| ΔCBS | None | 137 | — | — | — |

**Table 3 Analysis of oligomeric state of TbGMPR WT and TbGMPRΔCBS.**

or guanine nucleotides to the allosteric regulatory site. The GMPR of *T. brucei* is distinguishable from those of the host animals by the presence of CBS domain[6,9]; therefore, the CBS domain might be a good therapeutic target for African trypanosomiasis. Further structural studies of TbGMPR complexed with other nucleotides or their synthetic analogs should be performed to identify the compounds that specifically inhibit TbGMPR.

## Methods

**Heterologous expression and purification of TbGMPR**. The coding region of the TbGMPR gene was amplified by PCR using *T. brucei* ILTat 1.4 genomic DNA as a template. All primers used in the present study are described in Supplementary Table 7. The PCR product was cloned between the *Nde*I and *Hind*III sites of the pET-22b(+) expression vector. The construct was further manipulated to remove the CBS domain by inverse PCR to generate a TbGMPRΔCBS expression vector. Site-directed mutagenesis was performed according to the manufacturer's protocol of the KOD Plus mutagenesis kit (TOYOBO, Japan) to obtain the expression constructs for the TbGMPR W115R and C318A mutants.

TbGMPR and its variant recombinant proteins were expressed heterologously in *E. coli* BL21(DE3) as fusion proteins with histidine-tags at the C termini. *E. coli* cells harboring each expression construct were cultured at 37°C in LB medium supplemented with 100 mg mL$^{-1}$ ampicillin and 1% glucose. The culture was supplemented with 1 mM isopropyl-$\beta$-thiogalactopyranoside when the optical density at 600 nm exceeded 0.6, and then further incubated at 18 °C for 3 days. The cells were harvested and resuspended in 50 mM Tris-HCl (pH 7.8) containing 0.5 M KCl and 50 mM imidazole, and subsequently disrupted by sonication. The following purification procedure was performed at 4 °C. The supernatant was loaded onto Ni Sepharose 6 Fast Flow (GE Healthcare Japan), and then the bound proteins were eluted with a stepwise gradient of imidazole of 100, 150, 200, 300, and 500 mM. The fractions eluted with 150–500 mM imidazole were pooled and dialyzed for 2 days against a buffer consisting of 50 mM Tris-HCl (pH 7.8), 0.5 M KCl, 0.5 M arginine-HCl, 10 mM EDTA, and 1 mM DTT. The dialysate was further purified by gel-filtration using a Hiprep 26/60 Sephacryl S-200HR column (GE Healthcare Japan) equilibrated with a buffer composed of 50 mM Tris-HCl (pH 7.8), 0.5 M KCl, 0.5 M arginine-HCl, and 10 mM EDTA. The concentrations of purified proteins were determined by measuring the absorbance at 280 nm and using extinction coefficients predicted from the amino acid sequence of each protein. The molecular masses of purified recombinant TbGMPR and TbGMPRΔCBS were confirmed by mass spectrometry as described below.

**Evaluation of the effect of purine nucleotides on TbGMPR activity**. The kinetic parameters of TbGMPR in the presence or absence of purine nucleotides were determined as follows: The kinetic assay was carried out at 35 °C in the reaction buffer containing 200 μM NADPH, 100 nM TbGMPR, and 0–10 mM nucleotide ligands. When investigated with Mg-GTP or -ATP, magnesium chloride was premixed with GTP or ATP at an equivalent concentration prior to each reaction, and the reaction was performed in the reaction buffer without EDTA. The initial velocity of TbGMPR in the presence of various concentrations of GMP substrate was measured as the consumption of NADPH by monitoring the absorption at 340 nm ($\varepsilon_{340} = 6220$ M$^{-1}$ cm$^{-1}$) using UV-2600 spectrophotometer (Shimadzu) or Eon microplate reader (BioTek, VT). The kinetics of TbGMPRΔCBS was determined as described above, except that the enzyme was used at 1 μM. The initial velocity data ($n = 3$) were fitted to the Hill equation (Eq. 1) by using IGOR Pro 6.3 (WaveMetrics, OR).

$$v_0 = \frac{V_{max}[S]^{n\text{Hill}}}{K_{0.5}^{n}\text{Hill} + [S]^{n\text{Hill}}} \tag{1}$$

**Ligand-binding analysis by fluorescence quenching**. Fluorescence spectra of the tryptophan residue of TbGMPR W115R were measured at 35 °C in reaction buffer. The concentrations of purine nucleotides were varied to determine the dissociation constant. TbGMPR concentration was set to 1 μM for investigating GTP and GMP, or 5 μM for investigating ATP. The fluorescence spectra from 310 to 400 nm at an excitation wavelength of 290 nm were recorded with a F-7000 Fluorescence Spectrophotometer (Hitachi High-Technologies). The maximum intensity data ($n = 3$) plotted against the nucleotide concentrations were fitted to the Hill equation as described above.

**Crystallization and X-ray structure determination**. All crystallization experiments were performed on 24-well Cryschem plates (Hampton Research) by the sitting-drop vapor-diffusion method. The crystal of TbGMPR C318A apo-form (C318A apo) was obtained as follows: The protein was prepared at 10.2 mg mL$^{-1}$ in 50 mM Tris-HCl (pH 7.8) containing 0.5 M KCl, 10 mM EDTA, and 0.5 M arginine-HCl, then mixed with an equal volume of reservoir solution consisting of 0.1 M imidazole-HCl (pH 8.0), 3% (w/v) PEG 3350, and 0.35 M Li$_2$SO$_4$ on crystallization plates. The crystallization mixture was equilibrated against the reservoir solution at 4 °C. A micro-seeding technique was used to obtain a crystal of size sufficient for measuring X-ray diffraction. The crystal was mounted with aqueous polymer glue containing 10% (wt/vol) polyvinyl alcohol (average polymerization degree, 4500) and 5% (vol/vol) glycerol according to the method described previously[23]; the crystal then was placed in the diffractometer and exposed to the humid air (88% relative humidity) at 4 °C. After freezing under a stream of cold (−173 °C) nitrogen gas, the crystal was subjected to diffraction data collection.

Co-crystallization of the TbGMPR C318A and GMP (C318A/GMP) was performed at 4 °C as follows: The protein was prepared at 5.8 mg mL$^{-1}$ in a buffer solution consisting of 0.1 M HEPES (pH 7.5), 0.75 M NaH$_2$PO$_4$, and 0.75 M KH$_2$PO$_4$ and containing up to 3 mM GMP. Crystals suitable for X-ray diffraction were obtained as described above but without use of the micro-seeding technique. TbGMPR complexed with GTP (TbGMPR/GTP) was co-crystallized at 20 °C as follows: The protein was prepared at 7.6 mg mL$^{-1}$ in a buffer solution consisting of 0.1 M HEPES (pH 7.5), 0.8 M NaH$_2$PO$_4$, and 0.8 M KH$_2$PO$_4$ and containing up to 1 mM GTP. Small crystals were obtained as described above, and subsequently grown using the micro-seeding technique. Crystals of C318A/GMP and TbGMPR/GTP were cryoprotected with 30% (vol/vol) glycerol before flash-freezing in liquid nitrogen; diffraction images then were collected from crystals maintained under a cold nitrogen-gas stream.

X-ray diffraction data were collected with 1.0-Å synchrotron radiation on beamline BL38B1 at SPring-8 (Harima, Japan) using a PILATUS3 6M detector (Dectris). Indexing, merging, and scaling of the collected diffraction data were performed using the XDS program package[24]. The structures of C318A apo, C318A/GMP, and TbGMPR/GTP were determined by using the data at 2.80-, 1.90-, and 2.50-Å resolution, respectively, by molecular replacement using the *Pseudomonas aeruginosa* IMPDH[25] (PDBID: 4DQW) as a search model with Phaser[26] and Molrep[27]. Iterative rounds of model building and refinement were carried out by a combination of COOT, PHENIX, and Refmac5 in the CCP4i software package[28–31]. The structure and the crystal packing were analyzed by PyMOL (http://www.pymol.org) and PISA[13], respectively. The statistics associated with the collection, processing, and refinement are summarized in Supplementary Table 1. The atomic coordinates and structure factors of C318A apo, C318A/GMP, and TbGMPR/GTP have been deposited in the Protein Data Bank under ID codes 6JL8, 6JIG, and 6LK4, respectively.

**Size-exclusion chromatography small-angle X-ray scattering (SEC-SAXS)**. The SEC-SAXS experiments were performed at beamline B21, Diamond Light Source (Didcot, UK), coupled with in-line size-exclusion chromatography. Protein samples were formulated at concentrations of 4 or 12 mg mL$^{-1}$ in a buffer solution consisting of 50 mM Tris-HCl (pH 7.8), 0.5 M KCl, 3 mM EDTA, and 1 mM DTT in the presence or absence of purine nucleotides (10 mM). Sample was loaded onto a Shodex KW403-4F column (4.6 mm ID × 300 mm), which was connected into Agilent 1200 HPLC system (Waters) at a flow rate of 0.16 mL min$^{-1}$. The sample separated on the size-exclusion chromatography was exposed to X-rays in a 1.6-mm diameter, 10-μm thick quartz capillary flow cell, followed by data collection every 3 s. It was confirmed on the size-exclusion chromatogram system at B21 that the injected sample has been three times diluted at the exposure point. X-ray was focused on the detector, PILATUS 2 M (Dectris), the beam size was 1 mm (horizontal) × 0.5 mm (vertical) at the sample position and 0.08 mm (horizontal) × 0.07 mm (vertical) at the focal point. The wavelength of X-ray was 1 Å and the sample-detector distance was 4 m. The measurement temperature was 20 °C. Further information about the system can be found here: https://www.diamond.ac.uk/Instruments/Soft-Condensed-Matter/small-angle/B21/description.html. Raw SAXS 2-D images were processed with the DAWN[32] processing pipeline at the beamline to produce normalized and integrated 1-D unsubstracted SAXS curves. The background subtraction, averaging of the data and determination of the structure parameters and the molecular mass were performed using the program ScÅtter[33]. On using ScÅtter, the molecular mass was derived from the volume of correlation, which was directly estimated from the subtracted 1-D scattering curve[34]. Data

collection and scattering-derived parameters are summarized in Supplementary Table 6. The oligomeric states of TbGMPR were calculated by OLIGOMER[35] with fitting to the form-factor files calculated by FFMAKER[35] based on the crystal structures of TbGMPR (PDBIDs: 6JL8 and 6JIG for the relaxed and twisted octamers, respectively); the tetramer model was generated by dividing the twisted octamer structure (6JIG) in half.

**Mass spectrometry under denaturing conditions**. The purified recombinant TbGMPR and TbGMPRΔCBS (at monomeric concentrations of 50.4 and 90.3 µM, respectively) were buffer-exchanged into 30% formic acid in 150 mM ammonium acetate by passing the proteins through a Bio-Spin 6 column (Bio-Rad). The proteins were analyzed immediately by nanoflow electrospray ionization mass spectrometry equipped with gold-coated glass capillaries made in-house and loaded with ~2–5 µL sample per analysis. Spectra were recorded on a SYNAPT G2-Si HDMS mass spectrometer (Waters) in positive ionization mode at 1.63 kV with a 150-V sampling cone voltage and source offset voltage, 0-V trap and transfer collision energy, and 2-mL min$^{-1}$ trap gas flow. The spectra were calibrated with 1 mg mL$^{-1}$ cesium iodide and analyzed using Mass Lynx software (Waters).

**Reporting summary**. Further information on research design is available in the Nature Research Reporting Summary linked to this article.

## Data availability

Coordinates and structure factors have been deposited in the Protein Data Bank under accession numbers 6JL8, 6JIG, and 6LK4. The source data underlying Figs. 1a–f, 2a–c, 5a, b and Supplementary Figs. 1, 2a, b, 3a, b, 4, 5a, b, 6, 10 and 11 are provided as a Source Data file. Other data are available from corresponding author upon reasonable request.

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

## Acknowledgements

We thank Drs. O. Ishibashi, T. Bessho, and T. Kodama for helpful discussions. This work was supported by the program Grants-in-Aid for Scientific Research of the Ministry of Education, Culture, Sports, Science and Technology of Japan (Grant Nos. 25660231, 25242046, and 17K19329 to T.I.), and Osaka Prefecture. The synchrotron radiation experiments for the crystal structure analysis were performed under the approval of the Japan Synchrotron Radiation Research Institute (JASRI) (Proposal Nos. 2015B2099, 2016B2561, 2016A2699, 2016B2699, 2017A2504, 2017B2504, 2017A2696, 2017B2696, 2018A2550, 2018B2550, 2018A2551, 2018B2551, 2019A2571, and 2019B2571). We thank Diamond Light Source (Didcot, UK) for allowing access to the BioSAXS beamline B21. All the SEC-SAXS works were done in the in-house beamtime under the approval of Diamond Light Source. This research was supported by Joint Research by Exploratory Research Center on Life and Living Systems (ExCELLS).

## Author contributions

A.I. designed and performed the kinetic experiments and SEC-SAXS analysis, and wrote the manuscript. Tet.O. provided support for the kinetic experiments, and wrote the manuscript. M.T. and T.K. performed the kinetic experiments. Tak.O. and H.M. performed protein crystallization and structure solution. Kat.I. and R.R. performed SEC-SAXS analysis. Ken.I. and S.U. performed mass spectrometry. B.K.K. prepared materials. S.N. provided support for protein crystallization and structure solution, and wrote the manuscript. T.I. managed the project, designed experiments, and wrote the manuscript, with contributions from all co-authors.

## Competing interests

The authors declare no competing interests.
