## [Peer Review File · Nature Communications]

Reviewers' comments:

Reviewer #1 (Remarks to the Author):

The manuscript by Imamura et al. describes a special mechanism of regulation for the GMP reductase enzyme from *Trypanosoma* and *Leishmania*. The GMP reductase of these organisms is exceptional because it contains a CBS/Bateman domain, in contrast to all other known GMP reductases that lack it.

The biochemical data presented is certainly not new, since it has been already reported for the highly similar *Leishmania* GMP reductase enzyme (Smith et al. 2016, *Mol Microbiol* 100(5), 824-40). However, this manuscript reports novel and interesting structural insights that might have important implications for the design of new therapeutic strategies. I find the manuscript certainly interesting but, nonetheless, I have some serious concerns that the authors should take into account:

- Appropriate controls must be performed to demonstrate that the affinity tag present at the N-terminus of the TbGMMPR recombinant protein does not disturb either the oligomerization state nor the catalytic activity of the enzyme.

This issue is especially important in this case, since the activity of the highly similar GMMPR enzyme from *Leishmania* has been reported to be seriously affected by the presence of a 6xHis affinity tag at the N-terminus (thesis from McGill University "Characterization of a novel *Leishmania* 5'-Guanosine monophosphate reductase" in year 2006). Indeed, the untagged version of this protein has been used in a recent publication from these authors (Smith et al. 2016, *Mol Microbiol* 100(5), 824-40).

.- The reaction buffer used for the enzyme kinetic experiments, as well as the buffer used for SEC-SAXS, contains 3 mM EDTA and no Magnesium ions are added, thereby, there are no Magnesium free in the solution in these experiments. The authors should then demonstrate that Magnesium is not required for the binding of ATP and GTP to the enzyme, which would be expected.

.- In this manuscript, the reported value for K_{cat} is $11.1 \pm 0.26 \text{ min}^{-1}$. However, previously published data reports a significantly higher value: 4.7 s^{-1} (Li et al. 2015. *Mol Microbiol* 97(5), 1006-1020). The authors should discuss where this discrepancy might come from.

.- How do the different TbGMMPR structures compare to the conformational switch that has been described for the structural homolog IMP dehydrogenase? (Labesse et al. 2013. *Structure* 21(6),975-985; Buey et al. 2017. *Sci Reports* 7, 2648; Anthony et al. 2017. *Mol Biol Cell* 28(20), 2600-2608; Fernandez-Justel et al. 2019. *J Mol Biol* 431(5), 956-969).

It might be also convenient also to homogenize the nomenclature of the conformations; this is "extended/compacted" or "expanded/collapsed", instead of "relaxed/twisted", although this is only a personal consideration.

.- The authors call "allosteric site" to the nucleotide binding site where GMP and GTP bind. Do they refer to the second canonical nucleotide-binding site of the Bateman domain? Please, comment and indicate appropriately. Furthermore, is the first canonical site of the Bateman domain of TbGMMPR functional? It would be convenient to determine whether GTP and GMP bind only to the second or to both canonical sites.

.-How do the authors explain that the SAXS-derived value of R_g for TbGMMPR-ATP is larger than that for TbGMMPR-GTP, TbGMMPR-GMP and even apoTbGMMPR. It is expected that this value will be significantly smaller for tetramers than for octamers. What are the Porod's volumes, $I(0)$ values and the corresponding estimated molecular weights from the SAXS experiments?

.- Can the enzyme kinetic data be shown by using plots: V_{max} vs nucleotide (GTP or ATP) concentration and/or K_m vs nucleotide concentration?

.- Can ATP inhibit/dissociate the GTP/GMP-activated TbGMPr octamers? And, the other way round, can GTP/GMP activate the ATP-inhibited octamers? How would these data correlate to the intracellular concentrations stated in the manuscript, i.e. 1.28 mM for GTP and 3.7 mM ATP? Additionally, what is the EC50 value for ATP? It is difficult to infer this value from Figure 1C.

.- Could the authors comment/hypothesize on the molecular mechanisms resulting in the different enzymatic activities of the twisted and relaxed conformations? Might be the finger domains (or equivalent structures) involved? Is the accessibility to the active site different for these conformations? Does the affinity of the substrate for the active site change? Would be this mechanism comparable to what has been recently found for IMP dehydrogenases?

.- PDB entry "5X80: Crystal Structure of GMP reductase from *Trypanosoma brucei* with guanosine 5'-triphosphate", released in 2018, is the same than 6IHV?

.- And some minor points:

-> Abstract: The role of the CBS domain has already been characterized in *Leishmania*. Thereby, the statement "the role of this domain is still uncharacterized" should be changed.

-> Discussion (page 21, line 274): *Ashbya gossypii* IMPDH exists as a tetramer in the absence of purine nucleotides and not as an octamer as stated in this manuscript.

Reviewer #2 (Remarks to the Author):

The manuscript is well written and all of the experiment have been excused well. The major shortcoming of the manuscript is that with the exception of the crystallography data which adds incrementally to the knowledge of purine metabolism in kinetoplastids, the manuscript is largely a repeat of the information that was published describing the GMPr from *Leishmania*.

Point-by-point Responses to the comments of Reviewer #1

Remarks to the Author:

The manuscript by Imamura et al. describes a special mechanism of regulation for the GMP reductase enzyme from *Trypanosoma* and *Leishmania*. The GMP reductase of these organisms is exceptional because it contains a CBS/Bateman domain, in contrast to all other known GMP reductases that lack it.

The biochemical data presented is certainly not new, since it has been already reported for the highly similar *Leishmania* GMP reductase enzyme (Smith et al. 2016, *Mol Microbiol* 100(5), 824-40). However, this manuscript reports novel and interesting structural insights that might have important implications for the design of new therapeutic strategies. I find the manuscript certainly interesting but, nonetheless, I have some serious concerns that the authors should take into account:

Comment 1:

Appropriate controls must be performed to demonstrate that the affinity tag present at the N-terminus of the TbGMPR recombinant protein does not disturb either the oligomerization state nor the catalytic activity of the enzyme. This issue is especially important in this case, since the activity of the highly similar GMPR enzyme from *Leishmania* has been reported to be seriously affected by the presence of a 6xHis affinity tag at the N-terminus (thesis from McGill University “Characterization of a novel *Leishmania* 5'-Guanosine monophosphate reductase” in year 2006). Indeed, the untagged version of this protein has been used in a recent publication from these authors (Smith et al. 2016, *Mol Microbiol* 100(5), 824-40).

Response:

We thank the reviewer for the detailed review of our manuscript and the constructive comments. In the present study, we produced a recombinant TbGMPr protein with a C-terminal His₆ tag using the pET22b expression system, and demonstrated that purine nucleotides control the enzyme activity of this protein through binding to the allosteric site. As the reviewer commented, we have been cautious regarding the N-terminal tagging of this enzyme because of the very low expression level of recombinant TbGMPr tagged with GST (GST-TbGMPr) at the N-terminus. TbGMPr produced by the removal of the GST tag, which was used in a previous study, showed Michaelis-Menten kinetics. (Bessho, Okada, et al., 2016, PLoS Negl Trop Dis. 10(1): e0004339). However, we reinvestigated the kinetics of this untagged TbGMPr, including the data at low concentrations of GMP substrate, which had not been measured in the previous study, and found that the initial velocities (v_0) against GMP concentrations shows a sigmoidal curve (Reference Figure 1, see below). These findings indicate that GMP induces a positive cooperative effect on TbGMPr, both with and without a terminal tag.

Reference Figure 1 | Kinetic analysis of untagged recombinant TbGMPr with varied concentrations of GMP substrate. Recombinant TbGMPr was prepared with pGEX 6P-1 expression and purification system (GE healthcare Japan) as described previously (REF), and was used for the experiment after removal of the terminal tag. The initial velocities at the various concentrations of GMP were determined as described in Methods section, except that the concentrations of TbGMPr and NADPH were 15 $\mu\text{g/mL}$ and 20 μM , respectively, and the reaction temperature was set at 25°C. Note that the data were fitted to the Hill equation, indicating that GMP has a positive cooperativity effect on the enzyme. The V_{max} , $K_{0.5}$ and n_{Hill} constant were calculated as $1.37 \pm 0.02 \text{ nM}\cdot\text{sec}^{-1}$, $16.4 \pm 0.5 \mu\text{M}$ and 2.38 ± 0.17 , respectively. Experiments were performed in triplicate; values are plotted as mean \pm s.d.

Comment 2:

The reaction buffer used for the enzyme kinetic experiments, as well as the buffer used for SEC-SAXS, contains 3 mM EDTA and no Magnesium ions are added, Thereby, there are no Magnesium free in the solution in these experiments. The authors should then demonstrate that Magnesium is not required for the binding of ATP and GTP to the enzyme, which would be expected.

Response:

The components of the buffer used in the kinetics and SEC-SAXS experiments are

derived from our previous studies of TbIMPDH (Bessho, et al., 2013, Parasitology. 140(6):735-45) and GST-TbGMPR (Bessho, Okada, et al., 2016, PLoS Negl Trop Dis. 10(1): e0004339). In accordance with the reviewer's comments, we analyzed the effect of Mg^{2+} ions on TbGMPR activity in the presence or absence of ligand nucleotides. As shown in Reference Figure 2, Mg^{2+} ions have minimal effects on enzyme activation and the inhibition by GTP and

Reference figure 2 | Regulative effects of purine nucleotides on TbGMPR in the presence of magnesium ions. TbGMPR activity was measured as described in Methods section in EDTA-free buffer with or without 1 mM $MgCl_2$. (a) TbGMPR activity at the various concentrations of GTP in the presence of 100 μM GMP substrate. (b) TbGMPR activity at the various concentrations of ATP in the presence of 1 mM GMP substrate. Experiments were performed in triplicate; values are plotted as mean \pm s.d.

ATP. The effect of each nucleotide at 10 mM, the concentration used in the SEC-SAXS experiments, reached a plateau either in the presence or absence of Mg^{2+} ions. In addition, we further demonstrated that Mg^{2+} ions were less effective on ligand nucleotide binding to TbGMPR W115R by fluorescence quenching assay (Reference Figure 3), suggesting that the absence of Mg^{2+} ions has little or no effect on our SEC-SAXS analysis performed here.

Reference Figure 3 | Fluorescence quenching assay of purine nucleotide binding to TbGMPR W115R in the presence of magnesium ion. The fluorescence emission spectra of TbGMPR W115R were measured in EDTA-free buffer containing 1 mM MgCl₂. GTP (a) and ATP (b) were used as ligands. Note that each quenching profile is similar to that observed in the experiment without magnesium ion. Experiments were performed in triplicate; values are plotted as mean ± s.d.

Comment 3:

In this manuscript, the reported value for k_{cat} is $11.1 \pm 0.26 \text{ min}^{-1}$. However, previously published data reports a significantly higher value: 4.7 s^{-1} (Li et al. 2015. Mol Microbiol 97(5), 1006-1020). The authors should discuss where this discrepancy might come from.

Response:

In the article above, the authors characterized *T. brucei* GMP synthetase, not reductase, and hence the k_{cat} value of $11.1 \pm 0.26 \text{ min}^{-1}$ is that for GMP synthetase. We carefully read through the article, but could not find any description of TbGMPR.

Comment 4:

How do the different TbGMPR structures compare to the conformational switch that has been described for the structural homolog IMP dehydrogenase? (Labesse et al. 2013. Structure 21(6),975-985; Buey et al. 2017. Sci Reports 7, 2648; Anthony et al. 2017. Mol Biol Cell 28(20), 2600-2608; Fernandez-Justel et al. 2019. J Mol Biol 431(5), 956-969).

It might be also convenient also to homogenize the nomenclature of the conformations; this is “extended/compacted” or “expanded/collapsed”, instead of “relaxed/twisted”, although this is only a personal consideration.

Response:

It is a good idea to compare the different TbGMPR structures that we determined here to the conformational switch that has been described for the structural homolog IMP dehydrogenase (IMPDH). Accordingly, in Discussion section we provide our consideration on the differences in the activity regulation and the subunit conformations between TbGMPR and IMPDHs, as described in Line 291 (see below).

(Line 291) “In terms of the structures, TbGMPR shows very high homologies to IMPDHs, a $(\beta/\alpha)_8$ barrel protein also composed of catalytic and CBS domains⁷. The activity regulation of IMPDHs is accompanied by their conformational changes, and the relationship between the activity and the conformation is extensively investigated to date¹⁵⁻¹⁷: In the presence of ATP, IMPDHs form an active octamer composed of two tetramers. Whereas, the allosteric binding of guanine nucleotides alters the conformation of the IMPDH octamers to a compact form, and inhibits the activity by the interaction of the finger domains, the motifs that are believed to be

essential for the catalysis. On the other hand, our present findings indicate that TbGMMPR activity is regulated in positive and negative by allosteric binding of guanine and adenine nucleotides to the CBS domain, respectively, that is assumed to involve the conformational changes distinct from those observed in IMPDHs; represented as the conversion of relaxed and twisted octamer conformations in the absence and presence of guanine nucleotides, respectively, and the dissociation of the octamer into two tetramers in the presence of ATP. Further investigation of the precise structure of the active center of TbGMMPR is necessary to extend the understanding of the allosteric regulation mechanism at structure levels.”

We observed the twisting of the octamer conformation in TbGMMPR with GMP/GTP, and this is more apparent than the changes in overall size. So, we would like to use the nomenclature of the conformations as “relaxed/twisted”.

Comment 5:

The authors call “allosteric site” to the nucleotide binding site where GMP and GTP bind. Do they refer to the second canonical nucleotide-binding site of the Bateman domain? Please, comment and indicate appropriately. Furthermore, is the first canonical site of the Bateman domain of TbGMMPR functional? It would be convenient to determine whether GTP and GMP bind only to the second or to both canonical sites.

Response:

In the present study, we observed only two binding sites for guanine nucleotides on

TbGMMPR. One is the active site in the catalytic domain, and is specific for the substrate GMP. Another is the allosteric site found at the cleft or the interface between the catalytic and CBS domains, where either the ligand GMP or GTP can bind. We have not observed a second canonical nucleotide-binding site, as the reviewer mentioned. Thus, GMP molecules are found in both the active and the allosteric sites in the crystal structure of the TbGMMPR C318A/GMP complex (Fig. 4a), whereas the TbGMMPR/GTP complex contains a GTP molecule only in the allosteric site (Supplementary Figure 7a,b). This allosteric site is shared by GMP and GTP, as clearly indicated by the superimposed crystal structures of TbGMMPR C318A/GMP and TbGMMPR/GTP (Supplementary Figure 7b). These findings, together with our kinetics and SEC-SAXS data, indicate that the allosteric site presented here is functional for the regulation of TbGMMPR activity and conformation through binding of a GMP/GTP molecule. Of course, we do not deny the possibility of existence of a second canonical nucleotide-binding site.

Comment 6:

How do the authors explain that the SAXS-derived value of R_g for TbGMMPR-ATP is larger than that for TbGMMPR-GTP, TbGMMPR-GMP and even apoTbGMMPR. It is expected that this value will be significantly smaller for tetramers than for octamers. What are the Porod's volumes, $I(0)$ values and the corresponding estimated molecular weights from the SAXS experiments?

Response:

As the reviewer mentioned, the R_g value will increase proportionally to the molecular weight if the molecule is a rigid sphere. However, proteins generally have several flexible regions, and therefore, the correlation between R_g value and molecular weight is sometimes lost, even if a protein has a globular structure. Furthermore, in our OLIGOMER analysis (Table 3), 60.8% of TbGMMPR/ATP was estimated to be tetramers with a prolate shape (derived from a structure generated by dividing the C318A/GMP octamer in half). This might result in higher R_g values in SEC-SAXS measurements of TbGMMPR/ATP. We have included the Porod volumes, $I(0)$ values, and the corresponding estimated molecular weights from the SEC-SAXS experiments as Supplementary Table 6 of the revised manuscript.

Comment 7:

Can the enzyme kinetic data be shown by using plots: V_{max} vs nucleotide (GTP or ATP) concentration and/or K_m vs nucleotide concentration?

Answer:

We additionally performed the kinetic analyses with GTP or ATP at various concentrations. The $K_{0.5}$ value was decreased by GTP, but increased by ATP in a concentration-dependent manner (Supplementary Figure 1a), while the increase of k_{cat} was observed only in the presence of GTP (Supplementary Figure 1b). As a consequence, catalytic efficacy ($k_{cat}/K_{0.5}$) of TbGMMPR was improved and worsened in the presence of GTP and ATP, respectively, in a concentration-dependent manner (Fig. 1b). Therefore, in the revised version of the manuscript, we added a description of this point to the Results section (lines 99 and 107, see below).

(Line 99) “The addition of GTP to the reaction mixture enhanced TbGMMP activity by decreasing the $K_{0.5}$ accompanied with the increase of the k_{cat} value in a concentration-dependent manner (Table 1, Fig. 1a and Supplementary Figure 1a,b).”

(Line 107) “The increase in the $K_{0.5}$ value was observed in an ATP concentration-dependent manner (Supplementary Figure 1a). The opposed effects of GTP and ATP on TbGMMP activity were clearly observed when the values of catalytic efficiency ($k_{cat}/K_{0.5}$) were plotted against each ligand concentration. Although the $k_{cat}/K_{0.5}$ value of TbGMMP in the absence of the ligands was determined as $0.913 \times 10^5 \text{ M}^{-1} \text{ min}^{-1}$, it was increased by the addition of 10 μM GTP, and finally showed $4.36 \times 10^5 \text{ M}^{-1} \text{ min}^{-1}$ with 1 mM GTP (Fig. 1b and Table 1). In contrary, the $k_{cat}/K_{0.5}$ value was decreased by the addition of ATP and reached to $0.111 \times 10^5 \text{ M}^{-1} \text{ min}^{-1}$ with 1 mM ATP (Fig. 1b and Table 1).”

Comment 8:

Can ATP inhibit/dissociate the GTP/GMP-activated TbGMMP octamers? And, the other way round, can GTP/GMP activate the ATP-inhibited octamers? How would these data correlate to the intracellular concentrations stated in the manuscript, i.e. 1.28 mM for GTP and 3.7 mM ATP?

Additionally, what is the EC50 value for ATP? It is difficult to infer this value from

Figure 1C.

Response:

The behavior of TbGMMPR in the presence of both ATP and GTP at various concentration ratios is a very important issue to know the nature of the enzyme under the physiological conditions. To address the reviewer's comment, we performed additional kinetic experiment employing various concentrations of GTP and ATP simultaneously as the ligands. As shown in Fig. 1a, single use of 1 mM ATP inhibited TbGMMPR activity, however, the inhibition was revoked when combined with GTP at 100 μ M or more (Fig. 1c). The n_{Hill} value in these conditions was calculated to be 1.03 ± 0.68 ; this was very close to the value in the presence of GTP alone rather than that without the ligands. In contrast, ATP was less effective to TbGMMPR activated by 1 mM GTP (Supplementary Figure 2).

Description about the EC_{50} values of the ligands was withdrawn in revised version of the manuscript because of the replacement of Figure 1b and 1c. Alternatively, we calculated the EC_{50} values from the initial velocities of TbGMMPR with or without magnesium ions (Reference Figure 2). The EC_{50} values for GTP and ATP were as summarized in a table below, showing that GTP is effective at lesser concentrations than ATP, though the substrate concentrations used were different between the assays of these two ligands. Furthermore, the dissociation constants determined here by fluorescence quenching experiments indicate that the affinity of GTP is 30-fold higher than that of ATP. These findings suggest that TbGMMPR in trypanosomes ordinarily forms active and twisted octamer under the physiological conditions, enabling to maintain the intracellular ATP at higher concentration than GTP, i.e. 3.7 mM for ATP and 1.28 mM for GTP. Considering the above, we added a description to the Discussion

Ligand	EC_{50} (μ M)	
	Mg (-)	Mg (+)
GTP	5.0 ± 1.3	23.2 ± 2.4
ATP	490 ± 77	957 ± 89

section (lines 324, see below).

(Line 324) “The intracellular concentrations of GTP and ATP, that are reported to be 1.28 and 3.7 mM, respectively, in *T. brucei* bloodstream forms during logarithmic phase¹⁶, are sufficient to bind to the allosteric site of TbGMMPR. Our kinetic study revealed that only 100 μ M GTP was enough to re-activate TbGMMPR inactivated by 1 mM ATP (Fig. 1c), and furthermore, the dissociation constants determined here by fluorescence quenching experiments indicate that the affinity of GTP is 30-fold higher than that of ATP. These findings suggest that TbGMMPR in trypanosomes ordinarily forms active and twisted octamer under the physiological conditions, enabling to maintain the intracellular ATP at higher concentration than GTP.”

Comment 9:

Could the authors comment/hypothesize on the molecular mechanisms resulting in the different enzymatic activities of the twisted and relaxed conformations? Might be the finger domains (or equivalent structures) involved? Is the accessibility to the active site different for these conformations? Does the affinity of the substrate for the active site change? Would be this mechanism comparable to what has been recently found for IMP dehydrogenases?

Response:

Unfortunately, our structural data presented here are insufficient to discuss the above issues. Our present study demonstrated that GTP and ATP bind to the allosteric site on

TbGMMPR to alter its affinity (or $K_{0.5}$ values) to GMP substrate in opposed manners. Therefore, it is likely that the purine nucleotide binding to the allosteric site of TbGMMPR induces a certain degree of conformational changes around the active site, though the precise structures of the active site with or without ligands are still elusive. We determined the structure of the active site of the C318A/GMP complex; however, a large portion of the active site of C318A-apo is in a disordered state. In addition, the structure of ATP-bound TbGMMPR has been still undetermined. Without these structure data, we would like to avoid inferring the molecular mechanisms underlying the conformational changes in any more detail than already described in the present manuscript. To overcome these points, we are planning to investigate the structures of TbGMMPR with point mutations that interfere with the regulation of activity by purine nucleotides.

Comment 10:

PDB entry “5X80: Crystal Structure of GMP reductase from *Trypanosoma brucei* with guanosine 5'-triphosphate”, released in 2018, is the same than 6IHV?

Response:

PDB entry 5X80 is the co-crystal structure of GTP and TbGMMPR W115R, which we used in fluorescence quenching analysis in our present study. However, as shown in our manuscript, we succeeded in determining the co-crystal structure of wild-type TbGMMPR, and referred to this as 6LK4 in our manuscript. The RMSD value between these structures is only 0.30 Å for the alpha carbon atoms. PDBID of TbGMMPR/GTP has

been updated from 6IHY to 6LK4 because of deferring the data to be opened.

And some minor points:

Comment 1:

Abstract: The role of the CBS domain has already been characterized in *Leishmania*.
Thereby, the statement “the role of this domain is still uncharacterized” should be changed.

Response:

In accordance with the reviewer’s comment, we revised this statement to “the role of this domain is not fully understood.”

Comment 2:

Discussion (page 21, line 274): *Ashbya gossypii* IMPDH exists as a tetramer in the absence of purine nucleotides and not as an octamer as stated in this manuscript.

Response:

We appreciate to the reviewer’s suggestion. The description has been deleted according to the revision of the manuscript.

Point-to-point Response to Reviewer #2

The manuscript is well written and all of the experiment have been excused well. The major shortcoming of the manuscript is that with the exception of the crystallography data which adds incrementally to the knowledge of purine metabolism in kinetoplastids, the manuscript is largely a repeat of the information that was published describing the GMPR from *Leishmania*.

Response:

We thank the reviewer for pointing out the significance of our crystallographic analysis of TbGMPR to the knowledge of purine metabolism in kinetoplastids. The CBS domain is commonly found in trypanosomatids but rarely in other species of GMPRs. We would like to further investigate the structural changes around the active site of TbGMPR, in parallel with the activity changes induced by purine nucleotide ligands and their analogs.

Reviewers' comments:

Reviewer #1 (Remarks to the Author):

I consider that Imamura et al. have done an excellent job in reviewing the manuscript, which has certainly improved. I apologize for my wrong concern (comment 3), regarding the K_{cat} values of GMPR that I confused with a previous report on GMPS.

I have several concerns that might hopefully help to further improve the manuscript:

.- Why GTP masks (or extinguishes) the positive cooperative effect of GMP? GMP binds to the same allosteric site that GTP and the authors state in the manuscript that both GMP and GTP activate GMPR.

.- GTP readily activates ATP-inhibited GMPR but ATP is less effective in inhibiting the GTP-activated GMPR. Indeed, this is expected from the relative $K(1/2)$ values of ATP and GTP. Why do not the authors use a much higher ATP concentration for this experiment?

.- Regarding my previous "comment 5": what the authors call "allosteric site" is indeed the second canonical nucleotide binding site of a CBS or Bateman domain. These sites are defined by the numbering of the absolutely conserved Asp residues that coordinate the hydroxyls of the ribose moiety of the nucleotides. These residues are Asp149 (canonical site 1) and Asp211 (canonical site 2) in TbGMPR. I would highly recommend to clearly state this in the text.

.- Regarding my previous "comment 6": the authors are right: "proteins generally have several flexible regions, and therefore, the correlation between R_g value and molecular weight is sometimes lost, even if a protein has a globular structure". Could they show the Kratky plots to further demonstrate this statement?

.- Why the EC_{50} values for ATP and GTP, as reported in the table of the "comment 8" are so different in the presence or absence of Mg ions? I am also not convinced with the EDTA buffer used and the concentrations of Mg-free in the solution. The authors show 2 plots in the "comment 2" demonstrating that the presence or absence of 1 mM $MgCl_2$ do not largely affect the catalytic activity and the regulation of LmGMPR. However, total concentrations of 1 mM $MgCl_2$ and 10 mM ATP will result in 1mM Mg-ATP with no Mg-free in the solution. Usually, one would need to add 11 mM total $MgCl_2$ concentration to have 10 mM MgATP and 1 mM Mg-free in solution. I believe that the results presented in the manuscript are reliable but the authors should clearly state this issue in the text.

.- This is just a minor point; in line 205 (page 13), the authors state: "the CBS domain in C318A/GMP was rotated by approximately 40° with respect to the catalytic domain". I guess they mean that the CBS domain is rotated 40° between the APO and the GMP structures?

Point-by-point Responses to the comments of Reviewer #1

Reviewers' comments:

Reviewer #1 (Remarks to the Author):

I consider that Imamura et al. have done an excellent job in reviewing the manuscript, which has certainly improved. I apologize for my wrong concern (comment 3), regarding the K_{cat} values of GMPR that I confused with a previous report on GMPS. I have several concerns that might hopefully help to further improve the manuscript:

Comment 1:

Why GTP masks (or extinguishes) the positive cooperative effect of GMP? GMP binds to the same allosteric site that GTP and the authors state in the manuscript that both GMP and GTP activate GMPR.

Response:

We appreciate reviewer's great concern of our manuscript for further improvement. As the reviewer commented, both GMP and GTP can act as allosteric activators of TbGMPR through binding to the same allosteric regulatory site (canonical site 2). However, the affinity of GTP to the allosteric site is higher than that of GMP, as observed in our fluorescence quenching assay, so that GMP cannot bind simultaneously to the allosteric site if GTP concentration is enough to bind to the same site. Even though, our description in the previous version of the manuscript might not be appropriate to express the issues above. To clarify, we remove the sentence (Line 102-103 of the previous manuscript) "..., indicating that the high concentration of GTP

masked or extinguished the positive cooperativity effect of GMP on TbGMPR” from the Result section, and add the description to the Discussion section, Line 288 (see below).

(Line 288) “The present crystallographic study of TbGMPR shows that both GMP and GTP share the canonical site 2 on the CBS domain, whereas nothing was observed on the canonical site 1 (Fig. 4, Supplementary Figure 9 and Supplementary Table 5.). However, the affinity of GTP to the canonical site 2 was higher than that of GMP as observed in the fluorescence quenching assay (Fig. 2). These findings indicate that, in the presence of sufficient concentration of GTP, GMP no longer activates TbGMPR since the canonical site 2 has been occupied by GTP. Consequently, the kinetics of TbGMPR in the presence of GTP shows a Michaelis-Menten-like profile with a decrease of the n_{Hill} value to approximately 1 (Fig. 1c and Table 1).”

Comment 2:

GTP readily activates ATP-inhibited GMPR but ATP is less effective in inhibiting the GTP-activated GMPR. Indeed, this is expected from the relative $K(1/2)$ values of ATP and GTP. Why do not the authors use a much higher ATP concentration for this experiment?

Response:

It was impossible to use more concentration of ATP in the presence of 1 mM GTP because of too much absorption at the wavelength to monitor the reaction. Thus, we examined the effects of ATP at the concentrations within the measurable range on the

GTP-activated TbGMMPR by decreasing the GTP concentration to 100 μ M which is the concentration enough to produce nearly maximum activation. As the result, the addition of 3 mM ATP inhibited the 100 μ M GTP-activated TbGMMPR, while 1 mM ATP showed only a moderate effect. These data indicate that TbGMMPR is reversibly regulated between the activated and inhibited state by binding of GTP and ATP, respectively, and this state transition is dependent to the concentration ratio of these nucleotide ligands. The data are provided as Supplementary Figure 3a. We described the issue above in Results section of the revised version of the manuscript, as shown below.

(Line 136) “Inversely, the addition of 3 mM ATP inhibited the 100 μ M GTP-activated TbGMMPR, while 1 mM ATP showed only a moderate effect (Supplementary Figure 3a). ATP was less effective to 1 mM GTP-activated TbGMMPR (Supplementary Figure 3b). These data indicate that TbGMMPR is reversibly regulated between the activated and inhibited state by binding of GTP and ATP, respectively, and this state transition is dependent to the concentration ratio of these nucleotide ligands.”

Comment 3:

Regarding my previous “comment 5”: what the authors call “allosteric site” is indeed the second canonical nucleotide binding site of a CBS or Bateman domain. These sites are defined by the numbering of the absolutely conserved Asp residues that coordinate the hydroxyls of the ribose moiety of the nucleotides. These residues are Asp149 (canonical site 1) and Asp211 (canonical site 2) in TbGMMPR. I would highly recommend to clearly state this in the text.

Response:

According to reviewer's suggestion, we additionally mentioned in Discussion section about that both GMP and GTP can bind to the same allosteric regulatory site at Asp211 in TbGMMPR, and this site is corresponding to the one that referred to as canonical binding site 2 for the regulative enzymes through the purine nucleotide binding to their CBS domains. In the revised version of the manuscript, we actually described as shown below.

(Line 284) “In general, the regulative enzymes with CBS domains possess purine nucleotide binding sites in the domain, and these sites are referred to as canonical binding sites defined by the numbering of the highly conserved Asp residues that interact with the ribose moiety of the nucleotides^{12,14,15}. These Asp residues in TbGMMPR are Asp149 and Asp211, corresponding to the canonical binding site 1 and 2, respectively. The present crystallographic study of TbGMMPR shows that both GMP and GTP share the canonical site 2 on the CBS domain, whereas nothing was observed on the canonical site 1 (Fig. 4, Supplementary Figure 9 and Supplementary Table 5.)”

Comment 4:

Regarding my previous “comment 6”: the authors are right: “proteins generally have several flexible regions, and therefore, the correlation between Rg value and molecular weight is sometimes lost, even if a protein has a globular structure”. Could they show the Kratky plots to further demonstrate this statement?

Response:

In response to the suggestion, we have performed the Kratky plots of TbGMMPR with and without ligands. Consequently, the plots clearly showed an increase in the flexible structure only in case of ATP-binding TbGMMPR. Thus, we added these results in Results section of the revised version of the manuscript as shown below.

(Line 241) “Kratky plots of TbGMMPR exhibited a peak at Guinier-Kratky point in all measurements, but except for ATP, indicating that TbGMMPR substantially adopts the globular conformation (Fig. 5b). However, the plot of TbGMMPR in the presence of ATP showed a peak-shift from Guinier-Kratky point, and had no valley and diverged tailing at higher $q \times R_g$ region. These observations signify that ATP-binding results in an increase of the flexible structure in TbGMMPR, leading the subunit conformation into a less compact form.”

Comment 5:

Why the EC50 values for ATP and GTP, as reported in the table of the “comment 8” are so different in the presence or absence of Mg ions? I am also not convinced with the EDTA buffer used and the concentrations of Mg-free in the solution. The authors show 2 plots in the “comment 2” demonstrating that the presence or absence of 1 mM MgCl₂ do not largely affect the catalytic activity and the regulation of LmGMMPR. However, total concentrations of 1 mM MgCl₂ and 10 mM ATP will result in 1mM Mg-ATP with no Mgfree in the solution. Usually, one would need to add 11 mM total MgCl₂ concentration to have 10 mM MgATP and 1 mM Mg-free in solution. I believe that the results presented in the manuscript are reliable but the authors should clearly state this issue in the text.

Response:

We appreciate reviewer's detailed advice to improve our study. We now realized that the experimental protocol performed in our response to "comment 2" was not appropriate to examine the effect of magnesium ions on the GTP/ATP binding to the enzyme. In the physiological conditions, majority of GTP/ATP are considered to form complex molecule with magnesium ions, thus we focused on comparing the regulatory effects on TbGMPR between GTP/ATP and Mg-GTP/Mg-ATP. Prior to initiate the reaction, the mixture of equivalent moles of GTP/ATP and MgCl₂ was prepared, then used as described in Methods section. Under the present reaction conditions, we observed no changes on the regulatory effects of these nucleotides either in the presence or absence of magnesium ions. The EC₅₀ values were calculated as follows: GTP, 4.8; Mg-GTP, 3.2; ATP, 160; Mg-ATP, 152 (μM). These results are provided as Figure 1a and 1b, and are described in Results section in the revised version of the manuscript, as below.

(Line 92) "In the presence of GTP, the initial velocity of TbGMPR was upregulated in a concentration-dependent manner (Fig. 1a), and the EC₅₀ value was estimated to be 4.8 μM. In contrast, TbGMPR exhibited a decrease in its initial velocity in the presence of ATP with an EC₅₀ value of 160 μM (Fig. 1b). The effect of each triphosphate nucleotide was maintained when added as a magnesium complex, i.e. Mg-GTP or Mg-ATP (Fig. 1a,b). These results indicate that TbGMPR is positively and negatively regulated by GTP and ATP, respectively, with and without magnesium ions."

Comment 6:

This is just a minor point; in line 205 (page 13), the authors state: “the CBS domain in C318A/GMP was rotated by approximately 40° with respect to the catalytic domain”. I guess they mean that the CBS domain is rotated 40° between the APO and the GMP structures?

Response:

To make it be more comprehensive, we revised the description in the Results section as below, shown in blue with underline.

(Line 219) “However, the relative orientation of these domains was obviously different between C318A/GMP and C318A apo: the CBS domain is rotated approximately 40° between these two structures (Fig. 4a), a change that may be induced by an interaction between the base moiety of GMP and Arg93 (Fig. 4d,e).”